# Acute Ingestion of a Mixed Flavonoid and Caffeine Supplement Increases Energy Expenditure and Fat Oxidation in Adult Women: A Randomized, Crossover Clinical Trial

**DOI:** 10.3390/nu11112665

**Published:** 2019-11-05

**Authors:** David C. Nieman, Andy Simonson, Camila A. Sakaguchi, Wei Sha, Tondra Blevins, Jaina Hattabaugh, Martin Kohlmeier

**Affiliations:** 1Human Performance Laboratory, North Carolina Research Campus, Appalachian State University, Kannapolis, NC 28081, USA; simonsonaj@appstate.edu; 2Physical Therapy Department, Federal University of São Carlos, São Carlos 13565-905, Brazil; camilasakaguchi790@gmail.com; 3Bioinformatics Services Division, North Carolina Research Campus, University of North Carolina at Charlotte, Kannapolis, NC 28081, USA; wsha@uncc.edu; 4UNC Nutrition Research Institute, UNC-Chapel Hill, North Carolina Research Campus, 500 Laureate Way, Kannapolis, NC 28081, USA; tondra_blevins@unc.edu (T.B.); jaina_hattabaugh@unc.edu (J.H.); mkohlmeier@unc.edu (M.K.)

**Keywords:** flavonoids, metabolism, caffeine, energy expenditure, metabolic chamber

## Abstract

This randomized, double-blinded, crossover study measured the acute effect of ingesting a mixed flavonoid-caffeine (MFC) supplement compared to placebo (PL) on energy expenditure (EE) and fat oxidation (FATox) in a metabolic chamber with premenopausal women (*n* = 19, mean ± SD, age 30.7 ± 8.0 year, BMI 25.7 ± 3.4 kg/m^2^). The MFC supplement (658 mg flavonoids, split dose 8:30, 13:00) contained quercetin, green tea catechins, and anthocyanins from bilberry extract, and 214 mg caffeine. Participants were measured twice in a metabolic chamber for a day, four weeks apart, with outcomes including 22 h EE (8:30–6:30), substrate utilization from the respiratory quotient (RQ), plasma caffeine levels (16:00), and genotyping for the single-nucleotide polymorphism (SNP) rs762551. Areas under the curve (AUC) for metabolic data from the MFC and PL trials were calculated using the trapezoid rule, with a mixed linear model (GLM) used to evaluate the overall treatment effect. The 22 h oxygen consumption and EE were significantly higher with MFC than PL (1582 ± 143, 1535 ± 154 kcal/day, respectively, *p* = 0.003, trial difference of 46.4 ± 57.8 kcal/day). FATox trended higher for MFC when evaluated using GLM (99.2 ± 14.0, 92.4 ± 14.4 g/22 h, *p* = 0.054). Plasma caffeine levels were significantly higher in the MFC versus PL trial (5031 ± 289, 276 ± 323 ng/mL, respectively, *p* < 0.001). Trial differences for 22 h EE and plasma caffeine were unrelated after controlling for age and body mass (*r* = −0.249, *p* = 0.139), and not different for participants with the homozygous allele 1, A/A, compared to C/A and C/C (*p* = 0.50 and 0.56, respectively). In conclusion, EE was higher for MFC compared to PL, and similar to effects estimated from previous trials using caffeine alone. A small effect of the MFC on FATox was measured, in contrast to inconsistent findings previously reported for this caffeine dose. The trial variance for 22 h EE was not significantly related to the variance in plasma caffeine levels or CYP1A2*1F allele carriers and non-carriers.

## 1. Introduction

Long-term intake of green tea catechin-caffeine beverages has been linked to a small, positive effect on weight loss and weight maintenance [1,2]. Acute ingestion of catechin-caffeine mixtures may transiently increase energy expenditure and fat oxidation by inhibiting or stimulating several enzymes including catechol O-methyltransferase (COMT), phosphodiesterase, and hormone-sensitive lipase, and activating brown adipose tissue metabolic activity [1,2,3,4,5,6].

Five human studies (*n* = 66 males, *n* = 16 females) conducted in metabolic chambers used caffeine doses of 150 to 600 mg with varying amounts of green tea catechins and epigallocatechin-3-gallate (EGCG) (240 to 1200 mg) [7,8,9,10,11]. Increases in 24 h energy expenditure (24 h EE) with catechin-caffeine mixtures varied from 2% to 8% and were related more or entirely to the caffeine portion of the supplement. Rudelle et al. [10] proposed that the increase in 24 h EE with acute caffeine ingestion (with or without catechins) could be estimated by this equation: ((mg caffeine × 0.12) + 43.2 kilocalories).

The plasma caffeine response to caffeine intake may help explain the high inter-individual variance in energy expenditure changes, but this has not yet been measured. The linkage between caffeine intake, plasma caffeine levels, and variance in 24 h EE responses may also be influenced by the single-nucleotide polymorphism (SNP) rs762551 that encodes the cytochrome P450 1A2 (*CYP1A2)*1F* allele of the *CYP1A2* gene that produces the enzyme primarily responsible for the metabolism of caffeine [12]. An A to C substitution at position 163 (rs762551) in the *CYP1A2* gene decreases enzyme inducibility. Carriers of the C allele that is found in 54% of the population (163A/C and 163C/C genotypes, *CYP1A2*1F*) metabolize caffeine more slowly than individuals homozygous for the 163A/A allele (*CYP1A2*1A*).

Ingestion of catechin-caffeine mixtures may increase fat oxidation better than caffeine alone, but this finding has not been consistently supported in the few studies available. Two of the five studies showed that relative to placebo, catechin-caffeine mixtures stimulated fat oxidation above levels linked to caffeine alone [7,8], with three studies showing null effects [9,10,11]. High inter-individual variation was a common finding across these studies, with Rudelle et al. [10] reporting that six of 31 study participants did not respond with increased 24 h EE to the supplement dose (300 mg caffeine, 540 mg catechins, and 282 mg EGCG).

Acute ingestion of other types of flavonoids including flavonols and anthocyanins may influence energy expenditure and fat oxidation, but few metabolic chamber-based studies with human participants have been published [13,14,15,16]. Data from three prospective cohort studies involving 124,086 men and women showed that higher intake of foods and beverages rich in flavonols (e.g., quercetin from apples and onions), anthocyanins (e.g., from blueberries), flavan-3-ols (e.g., catechins from tea), and flavonoid polymers (e.g., proanthocyanidins and theaflavins from tea, apples) was inversely associated with weight change over four-year time intervals [17]. One study showed that quercetin transiently increased energy expenditure in mice [18], but this was not duplicated in a small pilot study with human participants ingesting 150 mg quercetin [16]. Acute ingestion of bilberry extract reduced postprandial increases in glucose by decreasing rates of carbohydrate absorption and may therefore have influenced fat oxidation [19]. Blackberries contain flavonols, anthocyanins, and flavan-3-ols, and high intake (600 g/day) for 7 days was linked to increased fat oxidation in overweight/obese males fed a high-fat diet relative to gelatin [15].

Acute ingestion of a mixed flavonoid-caffeine (MFC) supplement containing green tea extract, quercetin, and bilberry anthocyanins and proanthocyanidins has the potential to increase energy expenditure and fat oxidation above levels predicted for caffeine alone. The MFC supplement (678 mg flavonoids in four capsules) used in this study contained quercetin (200 mg), green tea catechins (368 mg, 180 mg EGCG), and anthocyanins (128 mg) from bilberry extract, and 214 mg caffeine. Based on the equation proposed by Rudelle et al. [10], the caffeine dose alone would be estimated to increase 24 h EE by 69 kcal/day. Male participant numbers in metabolic chamber-based studies have been low (*n* = 10 to 15 per study), and only one study included females (*n* = 16) [10]. Study designs and dosing regimens have varied considerably, with two of the five studies including a 3 day supplementation period with measurements only on the third day [8,10].

The purpose of this study was to measure the effect of ingesting the MFC supplement (four capsules, split between breakfast and lunch) compared to placebo on energy expenditure and fat oxidation in a metabolic chamber with healthy, premenopausal women. To reduce the potential influence of confounding factors, participants during both chamber visits (random order) followed the exact same schedule and were kept in an eucaloric state. Plasma caffeine levels were measured three hours after the second supplement dose, and genotyping for the single-nucleotide polymorphism (SNP) rs762551 encoding the *CYP1A2*1F* allele of the *CYP1A2* gene was conducted to determine whether these outcomes helped explain variance in energy expenditure responses to the placebo and MFC trials. We hypothesized that ingestion of the MFC supplement would increase energy expenditure and fat oxidation above levels measured with placebo and would be in part related to plasma caffeine levels and rs762551 genotype.

## 2. Materials and Methods

### 2.1. Study Participants

Using data generated previously in our metabolic chamber, a sample size calculation with 80% power revealed that 15–20 study participants would be needed to detect a 50 kcal difference in 24 h energy expenditure in a randomized, crossover clinical trial [20]. Study participants were recruited via mass advertisement. Inclusion criteria included: healthy, premenopausal females between the ages of 20–47 years; body mass index (BMI) between 18.5 and 33 kg/m^2^ with a stable weight during the past six months (±5.5 kg), some moderate-to-vigorous exercise but less than 300 min/week; willingness to maintain current body weight, habitual diet, and physical activity patterns during the study period; low-to-moderate caffeine intake (less than 3 servings per day of coffee, and/or caffeine-containing beverages); not taking herbal supplements or medications that could influence metabolism; willingness to report and maintain the normal schedule of hormonal therapy including oral contraceptive pills, hormonal intrauterine device (IUD), Nuva Ring, or DepoProvera injections; and willingness to follow all study protocols, including the ingestion of all provided foods while in the metabolic chamber. Study participants were excluded if they did not meet the inclusion criteria, or had any of the following: a history of cigarette smoking during the six months prior to the study; recent signs or symptoms of infection from the common cold or influenza; history of drug or alcohol abuse; thyroid hormone profile as measured through a pre-study blood sample that was clearly above or below the normal range; pregnant as determined by a urine pregnancy test during pre-study baseline testing; taking medications for or suffering from a medical condition that could impact results related to metabolism (e.g., thyroid disorders, diabetes, mental disorders such as anxiety or depression, heart disease, arthritis, cancer); current consumption (or unwilling to stop intake 2 weeks prior) of flavonoid supplements, or a heavy consumer of green tea (>3 cups per day); near-daily use of tablets containing caffeine (or unwillingness to stop 2 weeks prior to the study). Participants voluntarily signed the informed consent, and procedures were approved by the university Institutional Review Board. Trial Registration: ClinicalTrials.gov, U.S. National Institutes of Health, identifier: NCT03752125.

### 2.2. Research Design

This study utilized a randomized, double-blinded, cross-over design, comparing acute ingestion of the mixed flavonoid-caffeine supplement (MFC) to placebo control in healthy, premenopausal women. The study consisted of two 23 h study periods in the indirect room calorimeter at the University of North Carolina Chapel Hill Nutrition Research Institute (UNC NRI). The 23 h study periods in the metabolic chamber were 4 weeks apart to ensure that study participants were tested during the same phase of the menstrual cycle. Primary outcome measures were 22 h energy expenditure (22 h EE) (8:30 to 6:30), substrate utilization from the respiratory quotient (RQ), and physical activity counts. The 22 h time period was chosen to reduce perturbations in energy expenditure associated with transition time segments at the beginning and end of the chamber session.

#### 2.2.1. Supplement

As described previously [21], the MFC supplement and placebo capsules were prepared by Reoxcyn LLC (Pleasant Grove, UT, USA) and administered in a double-blinded manner. Supplement ingredients (US Patent 9,839,624) included the following (in 4 capsules) and provided 658 mg total monomeric flavonoids: 200 mg vitamin C (as ascorbyl palmitate) (Green Wave Ingredients, La Mirada, CA, USA), wild bilberry fruit extract with 128 mg anthocyanins (FutureCeuticals, Momence, IL, USA), green tea leaf extract with 368 mg total flavan-3-ols (Watson Industries, Inc., Pomona, CA, USA), 208 mg quercetin aglycone (Novel Ingredients, East Hanover, NJ, USA), 214 mg caffeine (Creative Compounds, Scott City, MO, USA), and 120 mg omega 3 fatty acids (Novotech Nutraceuticals, Ventura, CA, USA). Capsule fill ingredients and excipients included Nu-Flow 70R (from rice hulls), tapioca from cassava root, natural bamboo silica, and marshmallow root. Placebo capsules contained only the fill ingredients and excipients (without the active ingredients). The capsule contents were analyzed prior to the study for flavonoid content using high-performance liquid chromatography (HPLC) [21]. The flavonoid content was calculated as the sum of anthocyanins (measured as cyanidin-3-O-glucoside equivalents), quercetin, and flavan-3-ol compounds [epigallocatechin gallate (EGCG), epicatechin, epigallocatechin, and epicatechin gallate]. The monomeric flavonoid amounts listed do not add up to 658 mg due to differences in the molecular weight of mono- and polymeric forms.

#### 2.2.2. Pre-Study Baseline Testing

Eligibility was determined in the outpatient clinical suite at the UNC NRI. Body composition (fat mass and fat free mass (FFM)) was determined with dual energy X-ray absorptiometry (DXA) (GE Lunar iDXA; Milwaukee, WI). Body mass index (BMI, kg/m^2^) was calculated from measured height and weight. The resting metabolic rate (RMR) was estimated using the Mifflin-St. Jeor equation [22]. This estimated RMR was used to project dietary energy intake while in the metabolic chamber: RMR x physical activity level (PAL) of 1.3, with additional adjustments made during the chamber visits (details provided below). A small blood sample was collected to assess thyroid hormone status (Lab Corp, Burlington, NC, USA), and red blood cells were aliquoted and frozen for rs762551 genotyping. A urine pregnancy test was also performed.

#### 2.2.3. Indirect Calorimetry

The metabolic chamber at the UNC NRI in Kannapolis, NC, is an open-circuit, whole room indirect calorimeter. The CO_2_ and O_2_ analyzers are differential, with full scale readings set for 0–1%. Oxygen consumption (VO_2_), carbon dioxide production (VCO_2_), energy expenditure EE), and RQ were recorded each minute. Energy expenditure was calculated using an abbreviated Weir’s formula (VO_2_ × 3.941) + (VCO_2_ × 1.106), where VO_2_ is the volume of oxygen consumed in L/minute and VCO_2_ is the volume of carbon dioxide released in L/minute. RQ was calculated as VCO_2_/VO_2_. Substrate oxidation rates were calculated as follows: Fat oxidation rate (g/min) = (1.689 × VO_2_) − (1.689 × VCO_2_); carbohydrate oxidation rate (g/min) = (4.113 × VCO_2_) − (2.907 × VO_2_). Spontaneous physical activity was measured each minute using a total room microwave sensor that records a count for each second that motion is detected (Museum Technology Source Inc., Wilmington, MA, USA).

#### 2.2.4. Metabolic Chamber Protocol

Study participants arrived in an overnight fasted state at UNC NRI at 7:00, having avoided exercise for 24 h and caffeine and alcohol for at least eight hours. Participants were sealed in the chamber, with the recording of metabolic measurements starting at 8:00. Participants remained seated and awake throughout the day, with 2 min intervals scheduled hourly for standing, stretching, and washroom activities. MFC or placebo supplements were ingested at 8:30. And 13:00 (2 capsules each time). Breakfast (9:00), lunch (13:30), snack (16:00), and dinner (19:00) were served through an air-lock passage and consumed within 30 min of serving. A blood sample was obtained at 16:00 for measurement of plasma caffeine concentrations. Lights were turned off at 22:00, with bedtime set at 22:30. Study participants were awakened at 6:30 and allowed to move about the chamber to gather their belongings. At 7:15, study participants exited the chamber and were weighed. The 22 h EE was calculated from data collected from 8:30 to 6:30, with removal of the first and last 30 min transition segments.

#### 2.2.5. Metabolic Chamber Diet

Eucaloric diets were designed to provide approximately 35% fat, 50% carbohydrates and 20% protein, reflecting current recommendations for this population group. A low-flavonoid menu was designed using nutrient calculation and food management software (Nutribase: CyberSoft, Incorporated; Phoenix, AZ, USA), and consisted of rolled oats, white bread, peanut butter, milk (1% fat), lean ground turkey, brown rice, cottage cheese (1% fat), granola, dry roasted peanuts, boneless chicken breast, dry macaroni, and olive oil. No beverages or foods containing caffeine were served. The same foods were served at both chamber visits. A baseline menu for each subject was prepared based on an estimated energy expenditure (RMR × 1.3), and then modified (snack and dinner) according to measured energy expenditure data at 7 h.

### 2.3. Plasma Caffeine

Sample preparation was performed by protein precipitation of 50 μL plasma with 200 μL methanol containing 40 ng/mL internal standard caffeine-d9. Plasma caffeine concentration was analyzed by a Waters Acquity Ultra-Performance Liquid Chromatography (UPLC)-Quattro Premier XE Mass Spectrometry (Waters Corp., Milford, MA, USA). Chromatographic separation of caffeine and the internal standard caffeine-d9 was performed on a UPLC Ethylene Bridged Hybrid (BEH) C18 1.7 μm analytical column (2.1 × 100 mm, Waters Corp., Milford, MA, USA) with a gradient elution using mobile phase A water and mobile phase B acetonitrile, both containing 0.1% formic acid. Detection was performed using the mass spectrometer in the positive ion mode with electrospray ionization. The analysis was performed in the multiple reaction monitoring (MRM) mode. Instrument control and data acquisition were performed using Masslynx software package (Waters Corp., Milford, MA, USA).

### 2.4. rs762551 Genotyping with TaqMan Assay

DNA samples were extracted from 500 µl human blood in a preservation buffer using the QIAamp DNA blood mini kit. An amount of 20 ng of DNA was used as input for a genotyping reaction using the TaqMan SNP Genotyping Assay (ThermoFisher, Waltham, MA, USA). Predesigned primers were used to identify the three possible rs762551 genotypes (assay C_8881221_40). After qPCR reaction, the allele discrimination package from CFX Maestro Software from Bio-Rad (Hercules, CA, USA) was used to assign genotypes for each sample using three positive controls from Coriell Institute biobank for SNP calling (Camden, NJ, USA). DNA extraction and genotyping assays were performed at the Genomics Laboratory of the David H. Murdock Research Institute (Kannapolis, NC, USA).

### 2.5. Statistical Analyses

Data were analyzed using SAS 9.4 (SAS Institute Inc., Cary, NC, USA) and expressed as the mean ± SD. Two metabolic data curves, one for the MFC day, and one for the placebo control day, were generated for each subject with the x axis representing time (min), and the y axis representing the metabolic data (EE, VO2, VCO2, RQ, fat and carbohydrate oxidation, and activity counts). Areas under the curve (AUC) for each hour, defined time segments, and the entire 22 h period were calculated using the trapezoid rule in the EXPAND procedure in SAS (SAS Institute, Inc., Cary, NC, USA). A linear mixed model was used to evaluate the overall treatment (MFC and placebo) effect, time effect, and treatment × time interaction effect on log-transformed AUCs. Due to the crossover design, the visit effect and sequence effect were also examined. Paired *t*-tests on the log-transformed AUC areas were performed to compare the metabolic data between trials. Q-Q plot was used for normality check. The Benjamini–Hochberg method for false discovery rate correction in the MULTTEST procedure in SAS was used for multiple test corrections. Paired *t*-tests were used to compare trial differences for plasma caffeine levels. Independent *t*-tests were performed for selected outcome measures when comparing groups with genotypes C/A or C/C (*n* = 6) and genotype A/A (*n* = 11).

## 3. Results

The study participant flow diagram is shown in Figure 1. Of 23 female participants entered into the study, 20 successfully completed all phases of the study, with one dropped from the analysis due to a technical failure of the chamber (loss of temperature control). Study participants completing the study (*n* = 19) had these characteristics: age 30.7 ± 4.7 years, body mass 69.0 ± 9.0 kg, body mass index (BMI) 25.7 ± 3.4 kg/m^2^, and body fat 37.7 ± 6.6%.

Energy expenditures (kcal/h) for the MFC and placebo trials are compared in Figure 2 (treatment effect, *p* = 0.001). Total energy expenditure for the 22 h period for the MFC and placebo trials averaged 1582 ± 143 and 1535 ± 154 kcal, respectively (mean trial difference, 46.4 ± 57.8 kcal) (*p* = 0.003). Table 1 summarizes the metabolic data for each defined time segment. Activity counts and the respiratory quotient (RQ) were not significantly different between trials or during each segment (Table 1). Energy expenditure and oxygen consumption for the MFC trial were significantly higher than placebo (*p* < 0.001) when AUCs were compared as a whole and for each defined segment except during sleep.

Figure 3 shows the energy expenditure trial difference for each study participant. Five of 19 participants responded to the MFC supplement with lower energy expenditure compared to placebo (i.e., non-responders). For the responders, the trial difference was 70.9 ± 42.1 kcal. Age, BMI, and body fatness were not significantly different between responders and non-responders. Color-coded bars were used to represent genotyping for the SNP rs762551 (Figure 3). The trial difference for energy expenditure between those with allele 1, A/A, and those with C/A or C/C, was not different (51.4 ± 42.8 and 31.3 ± 82.1 kcal/22 h, respectively, *p* = 0.50).

Figure 4 compares fat oxidation rates on an hourly basis between MFC and placebo trials (treatment effect, *p* = 0.062). Fat oxidation for MFC trended higher than placebo across the defined time segments (*p* = 0.054), with no trial difference for carbohydrate oxidation (Table 1). Fat oxidation for the entire 22 h period for the MFC and placebo trials was 99.2 ± 14.0 and 92.4 ± 14.4 g/22 h, respectively (*p* = 0.081).

Plasma caffeine levels were significantly higher in the MFC than in the placebo trial (5031 ± 289, and 276 ± 323 ng/mL, respectively, *p* < 0.001). The trial differences for 22 h EE and plasma caffeine were unrelated after controlling for age and body mass (*r* = −0.249, *p* = 0.139). Figure 5 depicts plasma caffeine concentrations observed in samples drawn at 16:00 from those with one or two C alleles (*n* = 6) and with the homozygous genotype A/A (*n* = 11) groups in the placebo and MFC trials (mean ± SD). All had significantly higher caffeine concentrations in MFC trials than in placebo trials (*p* < 0.001), but trial contrasts were not different between genotype groups (*p* = 0.56).

## 4. Discussion

This study used a randomized, crossover, double-blinded, placebo-controlled design with 19 premenopausal women. The data showed that a supplement with a modest amount of caffeine (214 mg) and a flavonoid mixture (658 mg of green tea catechins, bilberry anthocyanins, and quercetin) was effective in increasing energy expenditure by 46 kcal/22 h and fat oxidation by 6.8 g/22 h. Trial differences in plasma caffeine levels and 22 h EE were unrelated after adjustment for age and body mass. Genotype for the rs762551 SNP did not explain the response variance in plasma caffeine and energy expenditure for the placebo and MFC trials.

The trial difference in energy expenditure (3%) given the MFC supplement dosing regimen is similar to what has been reported by other studies (2% to 8%) using metabolic chambers [7,8,9]. Our MFC supplement provided caffeine and EGCG amounts that were on the lower end of what has been used in previous studies [7,8,9,10]. Rudelle et al. [10] proposed that the increase in energy expenditure with caffeine-catechin supplements is related more to caffeine than the flavonoids. The formula provided by Rudelle et al. [10], projected an increased energy expenditure of 69 kcal/day for the 214 mg caffeine dose provided in this experiment. Our trial difference of 46 kcal/22 h is slightly below this level, and the smaller effect may in part be related to the female gender of the participants and to differences in study methodologies.

Participants varied widely in their metabolic response to the MFC supplement, and this variance could not be explained by body mass, age, spontaneous activity, plasma caffeine levels, or rs762551 SNP genotype. Heterogeneity in the metabolic response to caffeine and green tea EGCG supplementation is a common finding, but the underlying factors are not well understood [3,7,8,10,23]. After ingestion, caffeine is rapidly absorbed, peaking in the blood compartment after 60 min, with stimulation of the sympathetic nervous system (SNS) [24]. Caffeine-induced SNS activation, inhibition of phosphodiesterase (PDE), adenosine receptor antagonization, and stimulation of 5’adenosine monophosphate-activated protein kinase (AMPK) increase thermogenesis and fat oxidation [25]. The half-life of caffeine is approximately 5 h and is metabolized by liver cytochrome P450 (CYP) system enzymes into paraxanthine, theophylline, and theobromine metabolites [23]. This response, however, varies widely among individuals due in part to a polymorphism at the level of the *CYP1A2* and aryl hydrocarbon receptor (AHR) genes [12,24,25]. The SNP rs762551 encodes the *CYP1A2*1F* allele of the *CYP1A2* gene. However, rs762551(A) codes for the high inducibility form of the gene, with higher enzyme activity in response to behaviors such as heavy coffee consumption and smoking [12]. Participants in this study were all nonsmokers and habitually consumed less than three cups of coffee per day. Other polymorphisms that were not analyzed in this study, especially those encoding adenosine A1 and A2A receptors, may modify caffeine’s effect on anxiety, arousal, and sleep impairment [12]. Caffeine also has effects on appetite and spontaneous physical activity depending on the context, but this study utilized a structured design to control for these potential confounders [26,27].

Epidemiological data indicate that chronic ingestion of flavonoids is associated with lower long-term body weight and fat gain [17], but few human intervention studies have evaluated the acute effect of flavonoid intake on fat oxidation [16,18]. Previous studies using low to high caffeine doses alone, or caffeine with EGCG have reported variable results regarding effects on fat oxidation [7,8,9,10,23]. Metabolomics-based investigations revealed that ingesting flavonoid-caffeine supplements for 1–12 weeks increased plasma levels of metabolites related to fat metabolism [28,29]. We hypothesized that the acute ingestion of the MFC supplement with a high level of flavonoids (658 mg) would augment fat oxidation compared to placebo. Our data support a trend towards increased fat oxidation (7.4%) with the MFC supplement, but we could not distinguish between independent effects of caffeine and flavonoids. The flavonoid mixture used in this study combined green tea catechins with bilberry anthocyanins and quercetin, and limited data suggest that each of these flavonoids on their own may play a role in augmenting fat oxidation [12,13,14,15,16]. Additional research is needed to measure the acute fat oxidation effect of each of these flavonoids separately at low, medium, and high doses compared to caffeine alone and placebo.

## 5. Conclusions

In summary, the findings of this study support the hypothesis that the MFC supplement increases energy expenditure and fat oxidation in premenopausal women. Consistent with data from previous studies, the modest amount of caffeine (214 mg) together with the high amounts of mixed flavonoids (658 mg) in the MFC supplement increased energy expenditure by 3%, with a small, variable effect on fat oxidation in the female study participants. As reported in other metabolic chamber-based investigations, the metabolic response to the MFC supplement compared to placebo varied widely. Our data indicate that the trial variance was not related to plasma caffeine levels or explained by classification as *CYP1A2*1F* allele carriers and non-carriers.

## Figures and Tables

**Figure 1 nutrients-11-02665-f001:**
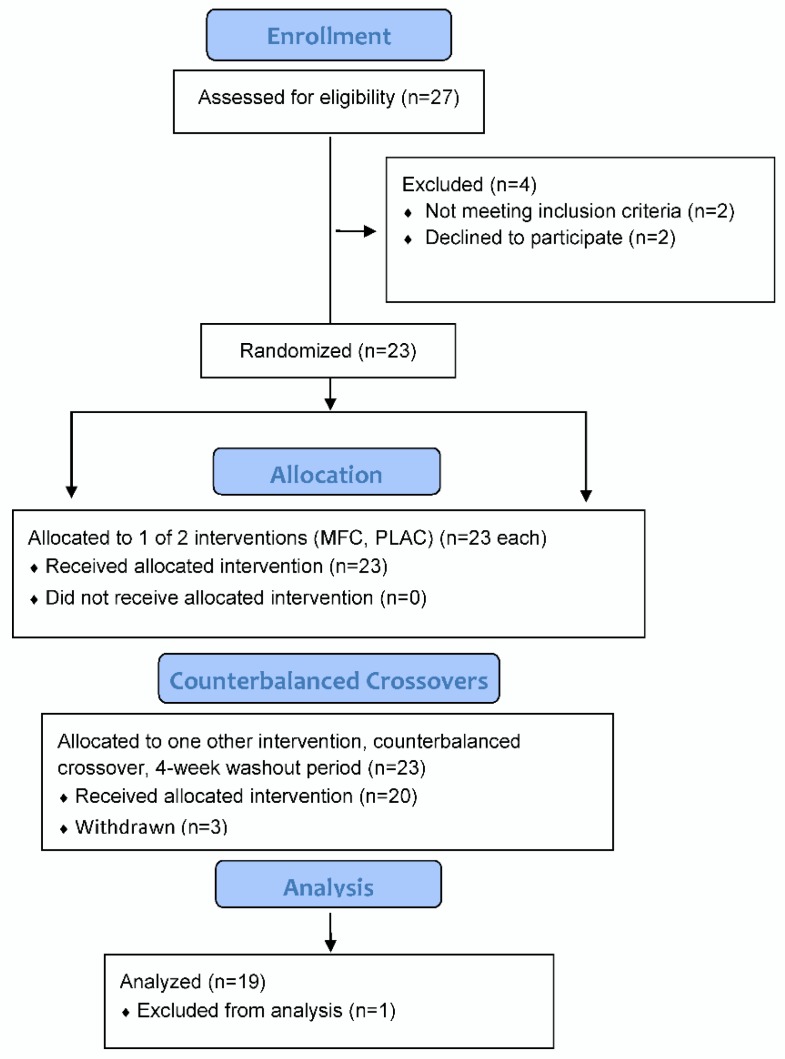
Study participant flow diagram. MFC = mixed flavonoid-caffeine trial; PLAC = placebo trial.

**Figure 2 nutrients-11-02665-f002:**
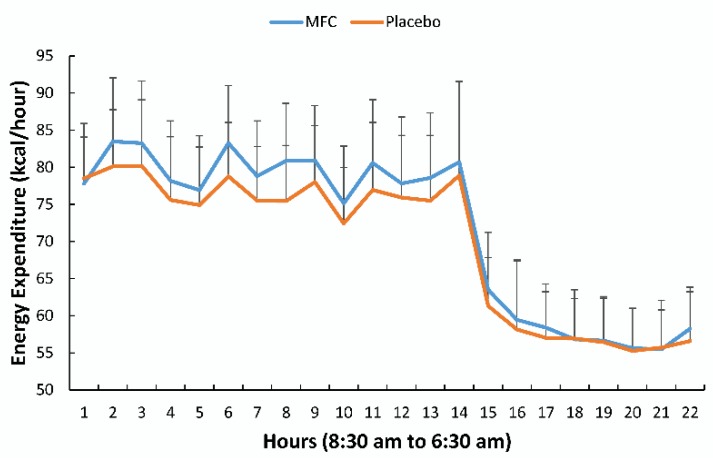
Energy expenditure (kcal/h) for the MFC and placebo trials (*n* = 19) (mean ± SD) (treatment effect, *p* = 0.001). Supplements were ingested at 8:30 (hour 1) and 13:00 (hour 5). Breakfast, lunch, and dinner were served at 9:00 (hour 1), 13:30 (hour 5), 18:30 (hour 10), respectively. Lights were turned off for sleep at 22:30 (hour 14).

**Figure 3 nutrients-11-02665-f003:**
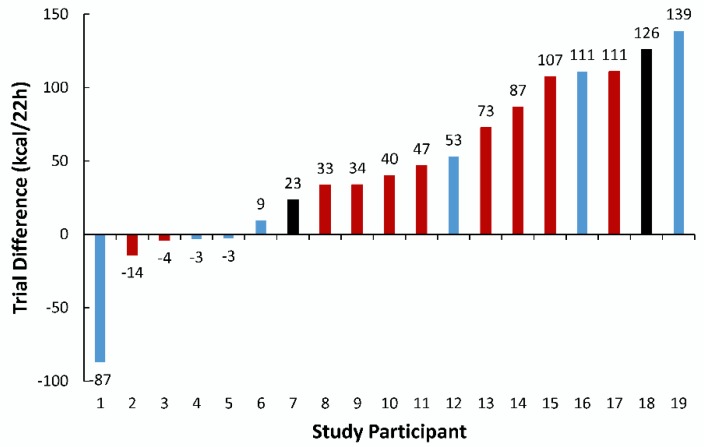
Energy expenditure (kcal/22 h) trial difference (MFC minus placebo) for each study participant. Data are depicted in ascending order, with arbitrary identification numbers for study participants. Red bars indicate study participants with the homozygous genotype A/A of the single-nucleotide polymorphism (SNP) rs762551. Blue bars indicate study participants with the genotypes C/A and C/C (just one participant, #4). The black bars indicate two participants for genotypes that could not be determined. The trial difference in energy expenditure between those with genotype A/A, and those with genotypes C/A or C/C was not statistically significant (51.4 ± 42.8 and 31.3 ± 82.1 kcal/22 h, respectively, *p* = 0.50).

**Figure 4 nutrients-11-02665-f004:**
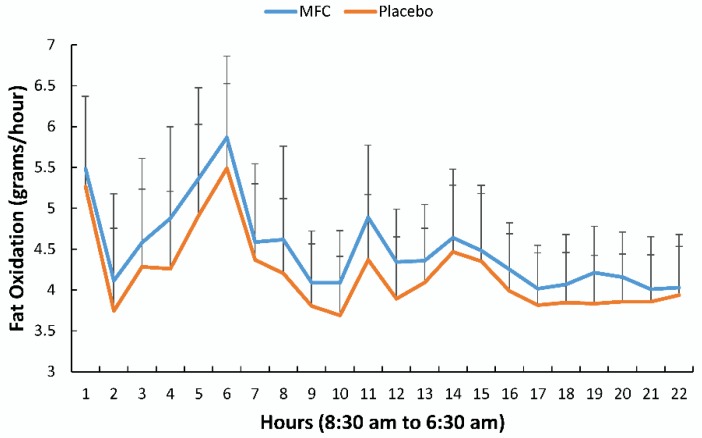
Fat oxidation (grams/hour) for the MFC and placebo trials (*n* = 19) (mean ± SD) (treatment effect, *p* = 0.062). Supplements were ingested at 8:30 (hour 1) and 13:00 (hour 5). Breakfast, lunch, and dinner were served at 9:00 (hour 1), 13:30 (hour 5), and 18:30 (hour 10), respectively. Lights were turned off for sleep at 22:30 (hour 14).

**Figure 5 nutrients-11-02665-f005:**
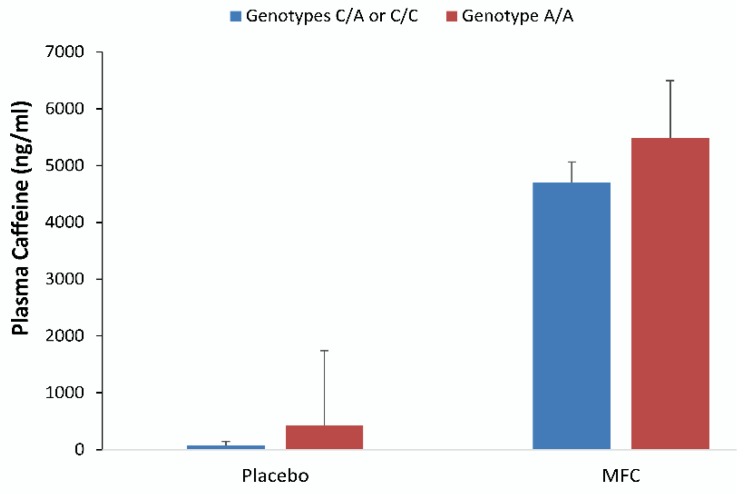
Plasma caffeine concentrations measured in samples drawn at 16:00 for those with genotypes C/A or C/C (*n* = 6), and genotype A/A (*n* = 11) in the placebo and MFC trials (mean ± SD). Both groups had significantly higher caffeine concentrations when comparing MFC and placebo trials (*p* < 0.001). The difference in trial contrasts between genotype groups was not statistically significant (*p* = 0.56).

**Table 1 nutrients-11-02665-t001:** Average metabolic data (expressed per minute) for the mixed flavonoid-caffeine trial (MFC) and placebo trials by each time segment (*n* = 19) (mean ± SD).

Variable	8:30–13:00 1st to 2nd Supplement	13:01–16:00 2nd Suppl. to Snack	16:01–19:00 Snack to Dinner	19:01–22:00 Dinner to Bedtime	22:01–6:30 Sleep	Treatment Effect (*p*-Value)
EE (kcal/min)						
MFC Placebo	1.35 ± 0.12 * 1.31 ± 0.13	1.33 ± 0.12 * 1.27 ± 0.12	1.32 ± 0.12 * 1.27 ± 0.13	1.32 ± 0.15 * 1.27 ± 0.14	0.99 ± 0.10 0.98 ± 0.11	0.0003
VO_2_ (L/min)						
MFC Placebo	0.277 ± 0.025 * 0.268 ± 0.026	0.275 ± 0.025 * 0.262 ± 0.024	0.271 ± 0.025 * 0.260 ± 0.026	0.272 ± 0.030 * 0.261 ± 0.029	0.206 ± 0.021 0.203 ± 0.021	0.0002
VCO_2_ (L/min)						
MFC Placebo	0.229 ± 0.021 0.225 ± 0.023	0.224 ± 0.020 * 0.215 ± 0.020	0.228 ± 0.021 * 0.221 ± 0.023	0.227 ± 0.027 0.221 ± 0.027	0.164 ± 0.019 0.163 ± 0.019	0.0123
RQ (VCO_2_/VO_2_)						
MFC Placebo	0.830 ± 0.030 0.838 ± 0.031	0.819 ± 0.026 0.823 ± 0.026	0.844 ± 0.020 0.851 ± 0.026	0.836 ± 0.024 0.845 ± 0.026	0.796 ± 0.023 0.804 ± 0.024	0.2349
C_Ox_ (g/min)						
MFC Placebo	0.139 ± 0.034 0.144 ± 0.037	0.125 ± 0.028 0.123 ± 0.028	0.151 ± 0.024 0.153 ± 0.032	0.143 ± 0.031 0.148 ± 0.034	0.076 ± 0.023 0.083 ± 0.025	0.6631
F_Ox_ (g/min)						
MFC Placebo	0.079 ± 0.017 0.074 ± 0.016	0.083 ± 0.015 0.079 ± 0.015	0.070 ± 0.012 0.066 ± 0.013	0.074 ± 0.012 0.069 ± 0.011	0.069 ± 0.008 0.066 ± 0.009	0.0539
Activity (counts/min)						
MFC Placebo	3.20 ± 1.05 3.18 ± 0.85	2.66 ± 0.78 2.50 ± 0.45	3.18 ± 0.86 3.34 ± 1.34	2.54 ± 0.93 2.40 ± 0.50	0.53 ± 0.26 0.51 ± 0.25	0.6662

* = fdr *p*-value ≤ 0.05, trial difference at time segment. EE = energy expenditure; VO_2_ = oxygen consumption; VCO_2_ = carbon dioxide production; RQ = respiratory quotient; C_Ox_ = carbohydrate oxidation; F_Ox_ = fat oxidation.

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
