# Peer review of "Acute Ingestion of a Mixed Flavonoid and Caffeine Supplement Increases Energy Expenditure and Fat Oxidation in Adult Women: A Randomized, Crossover Clinical Trial"

_nutrients, 2019, doi:10.3390/nu11112665_

Round 1

Reviewer 1 Report

The results of the study are interesting overall, though more information is needed about methods and design choices.  The results are nicely presented.  

Specific suggestions:

1) Caffeine and flavonoid intake do not appear to have been controlled during the days prior to the metabolic testing.  Since caffeine can induce cyp1A2, how might the lack of tight control of caffeine intake have affected the results?  Alternatively, if the intake of caffeine prior to testing was tightly controlled, please describe.  

2) Line 123: Volunteers were excluded if they had "near-daily use of tablets containing caffeine" - was there a specific use pattern that warranted exclusion?  "Near-daily" is a bit vague; please describe.

3) Lines 131-133: Menstrual cycles vary such that separating chamber runs by 4 weeks does not ensure that participants were tested during the same phase in the menstrual cycle.  95% CI for menstrual cycle length is about 23-32 days. 

4) Please provide a more detailed explanation of the supplement ingredients.  4a) Why was vitamin C included in the supplement?  4b) Why were omega 3 fatty acids included in the supplement?  4c) How might their inclusion have affected the results?   4d) The fill ingredients in the supplement and the placebo contained cassava root (which has been reported to contain flavonoids) and marshmallow root (which contains bioactives purported to have physiologic effects), thus it would be important to ensure that the amount of fill was the same in all capsules.  4e) Please explain the choice of fill ingredients.  4f) Were the amount of fill ingredients in every capsule exactly matched?  4g) How much fill was used?  4h) Were the placebo pills tested for flavonoid content?  

5) Figure 4 shows fat oxidation across the measurement period.  Please provide details about how the fat oxidation was calculated.  Indirect calorimetry data does not specifically provide that value; it must be calculated.  

Author Response

Manuscript ID   nutrients-619761

Type

Acute Ingestion of a Mixed Flavonoid and Caffeine Supplement Increases Energy Expenditure and Fat Oxidation in Adult Women: A Randomized, Crossover Clinical Trial

Authors

David Nieman * , Andrew Simonson , Camila Sakaguchi , Wei Sha , Tondra Blevins , Jaina Hattabaugh , Martin Kohlmeier

Reviewer #1:

In this RCT the acute effect of a mixed flavonoid-caffeine supplement on energy expenditure and fat oxidation was studied in pre-menopausal women. As hypothesized, the mixture increased energy expenditure and fat oxidation. The interindividual variation of the response was large, but was not explained by the plasma caffeine levels or the CYP1A2 genotype.

The number of subjects is a bit low, especially given the fact that one of the aims was to assess the effect of different genotypes of the CYP1A2 gene in caffeine plasma levels and energy expenditure/fat oxidation. Nevertheless the study was well-performed and the results are presented and discussed clearly.

Specific comments:

Was there a rationale for this specific combination of flavonoids or was it a commercially available supplement? How does the dose of caffeine in the supplement compare to habitual caffeine intake from coffee or caffeine-containing beverages?

RESPONSE:  Thank you for taking the time and effort to review our paper.

The mixed flavonoid-caffeine supplement was recently formulated (see methods section and patent number). Details are also provided in reference #21.  The supplement was formulated to take advantage of the potential synergism of consuming 3 classes of flavonoids with caffeine and nutrients that may enhance absorption and bioactivity. This supplement is recently commercially available. The average U.S. female adult consumes 150 mg caffeine per day, but the caffeine dose in this study was based on the published studies reviewed in the introduction.

What was the rationale to determine the influence of rs762551 and not other functional SNP's in the CYP1A2 gene? Is the caffeine metabolism higher or lower in carriers of the rs762551 AA genotype? Are the metabolites of caffeine active or inactive with respect to energy expenditure and fat oxidation?

RESPONSE:  We targeted this SNP because of the literature support reviewed in the paper by Nehlig (reference #12).  Here is the rationale from that paper, and this information has been added to the introduction of the paper (in response to your comment):

"The variability of CYP1A2 activity is largely explained by a gene polymorphism. An A to C substitution at position 163 (rs762551) in the CYP1A2 gene decreases enzyme inducibility as reflected by plasma or urinary caffeine-to-metabolite ratios after caffeine intake. The distribution of CYP1A2 activity in the population appears to be bimodal with the C variant being associated with slowing down of caffeine clearance. Carriers of the C allele, 54% of the population (163A/C and 163C/C genotypes; CYP1A2*1F), metabolize caffeine more slowly than individuals homozygous for the 163A/A allele (CYP1A2*1A) who are considered rapid caffeine metabolizers and represent 46% of the population."

The last sentence of the discussion should be modified. Firstly, it remains to be shown that the effect remains present and of the same magnitude with prolonged administration of the supplement. Secondly, no compensatory increase in energy intake should ensue.  

RESPONSE:  We deleted the last sentence (and accompanying reference) of the discussion in response to your comment.

Line 34: delete '/' after C/C.

RESPONSE:  Deleted as recommended.

Line 86: increase instead of increased

RESPONSE:  Corrected as recommended.

Line 111: caffeine-containing beverages

RESPONSE:  Corrected as recommended.

Line 175: given the caffeine half-life of 5 h, is a caffeine-abstention of 8 hours sufficient to prevent an influence of differences in prior caffeine consumption among participants?

RESPONSE: We cannot rule out the potential effect of the previous day's intake of caffeine on the metabolism of participants as they entered the chamber (8:00 am).  However, participants in this randomized, crossover trial operated as their own controls, and we feel that the effect of this potential confounder was small or nonexistent when evaluated across trials.

Reviewer 2 Report

In this RCT the acute effect of a mixed flavonoid-caffeine supplement on energy expenditure and fat oxidation was studied in pre-menopausal women. As hypothesized, the mixture increased energy expenditure and fat oxidation. The interindividual variation of the response was large, but was not explained by the plasma caffeine levels or the CYP1A2 genotype.

The number of subjects is a bit low, especially given the fact that one of the aims was to assess the effect of different genotypes of the CYP1A2 gene in caffeine plasma levels and energy expenditure/fat oxidation. Nevertheless the study was well-performed and the results are presented and discussed clearly.

Specific comments:

Was there a rationale for this specific combination of flavonoids or was it a commercially available supplement? How does the dose of caffeine in the supplement compare to habitual caffeine intake from coffee or caffeine-containing beverages?

What was the rationale to determine the influence of rs762551 and not other  functional SNP's in the CYP1A2 gene? Is the caffeine metabolism higher or lower in carriers of the rs762551 AA genotype? Are the metabolites of caffeine active or inactive with respect to energy expenditure and fat oxidation?

The last sentence of the discussion should be modified. Firstly, it remains to be shown that the effect remains present and of the same magnitude with prolonged administration of the supplement. Secondly, no compensatory increase in energy intake should ensue.  

Line 34: delete '/' after C/C.

Line 86: increase instead of increased

Line 111: caffeine-containing beverages

Line 175: given the caffeine half-life of 5 h, is a caffeine-abstention of 8 hours sufficient to prevent an influence of differences in prior caffeine consumption among participants?

Author Response

Reviewer #2:

In this RCT the acute effect of a mixed flavonoid-caffeine supplement on energy expenditure and fat oxidation was studied in pre-menopausal women. As hypothesized, the mixture increased energy expenditure and fat oxidation. The interindividual variation of the response was large, but was not explained by the plasma caffeine levels or the CYP1A2 genotype.

The number of subjects is a bit low, especially given the fact that one of the aims was to assess the effect of different genotypes of the CYP1A2 gene in caffeine plasma levels and energy expenditure/fat oxidation. Nevertheless the study was well-performed and the results are presented and discussed clearly.

Specific comments:

Was there a rationale for this specific combination of flavonoids or was it a commercially available supplement? How does the dose of caffeine in the supplement compare to habitual caffeine intake from coffee or caffeine-containing beverages?

RESPONSE:  Thank you for taking the time and effort to review our paper.

The mixed flavonoid-caffeine supplement was recently formulated (see methods section and patent number). Details are also provided in reference #21.  The supplement was formulated to take advantage of the potential synergism of consuming 3 classes of flavonoids with caffeine and nutrients that may enhance absorption and bioactivity. This supplement is recently commercially available. The average U.S. female adult consumes 150 mg caffeine per day, but the caffeine dose in this study was based on the published studies reviewed in the introduction.

What was the rationale to determine the influence of rs762551 and not other functional SNP's in the CYP1A2 gene? Is the caffeine metabolism higher or lower in carriers of the rs762551 AA genotype? Are the metabolites of caffeine active or inactive with respect to energy expenditure and fat oxidation?

RESPONSE:  We targeted this SNP because of the literature support reviewed in the paper by Nehlig (reference #12).  Here is the rationale from that paper, and this information has been added to the introduction of the paper (in response to your comment):

"The variability of CYP1A2 activity is largely explained by a gene polymorphism. An A to C substitution at position 163 (rs762551) in the CYP1A2 gene decreases enzyme inducibility as reflected by plasma or urinary caffeine-to-metabolite ratios after caffeine intake. The distribution of CYP1A2 activity in the population appears to be bimodal with the C variant being associated with slowing down of caffeine clearance. Carriers of the C allele, 54% of the population (163A/C and 163C/C genotypes; CYP1A2*1F), metabolize caffeine more slowly than individuals homozygous for the 163A/A allele (CYP1A2*1A) who are considered rapid caffeine metabolizers and represent 46% of the population."

The last sentence of the discussion should be modified. Firstly, it remains to be shown that the effect remains present and of the same magnitude with prolonged administration of the supplement. Secondly, no compensatory increase in energy intake should ensue.  

RESPONSE:  We deleted the last sentence (and accompanying reference) of the discussion in response to your comment.

Line 34: delete '/' after C/C.

RESPONSE:  Deleted as recommended.

Line 86: increase instead of increased

RESPONSE:  Corrected as recommended.

Line 111: caffeine-containing beverages

RESPONSE:  Corrected as recommended.

Line 175: given the caffeine half-life of 5 h, is a caffeine-abstention of 8 hours sufficient to prevent an influence of differences in prior caffeine consumption among participants?

RESPONSE: We cannot rule out the potential effect of the previous day's intake of caffeine on the metabolism of participants as they entered the chamber (8:00 am).  However, participants in this randomized, crossover trial operated as their own controls, and we feel that the effect of this potential confounder was small or nonexistent when evaluated across trials.

Reviewer 3 Report

General comments:

The authors conducted a randomized control trial on the effects of mixed flavonoid-caffeine supplement on energy expenditure readouts in women. The main finding is that the dietary intervention increases energy expenditure and lipid utilization, and that these effects are associated with caffeine. The paper is well written and the RCT has significant translational potential, but there are major concerns.

Major comments:

Page 2 lines 93-96, page 3 lines 127-35, page 4 lines 173-184: Please move some of the experimental design aspects (lines 92-95) to the appropriate methods section. The authors indicate that blood samples were drawn at ~3 hrs after the 2nd supplement dose, but it is stated that energy expenditure was measured continuously in the chambers. It is not clear what was did to the subjects and when. A figure with a time-line showing the experimental interventions should help better understand the design.

Page 4 lines 185-190, and page 5 lines 231-236: Though there is extensive description of the diet, there is no data on food intake and whether the MFC affected any indices of satiety. It is important to provide caloric intake data after each intervention and determine whether the intake is correlated with energy expenditure and other readouts.

Page 6 lines 239-253 and Figure 2: Overall, the study seems to lack mechanistic depth into why and how the MFC increased expenditure. For example, the data indicates that the MFC increased total energy expenditure but does not partition this into other components other than showing a lack of any changes in activity. Though Table 1 shows expenditure in different segments it’s not clear to me whether there is a specific diet effect. As the expenditure was monitored after an overnight fast, can you estimate diet-induced thermogenesis? This could provide some insights into the components of expenditure that are altered.

Page 3 lines 127-35, page 9 lines 308-309, and page 10 lines 340-346: A major limitation of the design is the use of a supplement that contains a complex mixture of flavonoids and caffeine and the inability to parse the individual effects. Though the authors briefly touched on this limitation (lines 340-343), I think this needs to be elaborated more. The authors estimate that the caffeine content of their diet would have increased expenditure by ~46 kcal vs the predicted ~69 kcal. This is a much larger ~50% differential but there is no clear discussion on this other than a vague mention of method differences. Majority of the discussion focusses on caffeine but it’s not clear whether the other components in MFC could potentially decrease expenditure and counteract the effects of caffeine.

Page 10 lines 355-356: The energy expenditure responses to MFC were only measured to 22 h. There is no data provided to indicate that the MFC would lead to sustained elevation in expenditure. In the absence of such data, the claim that the MFC could offer a long-term strategy for obesity is unfounded.

Author Response

Reviewer #3

The authors conducted a randomized control trial on the effects of mixed flavonoid-caffeine supplement on energy expenditure readouts in women. The main finding is that the dietary intervention increases energy expenditure and lipid utilization, and that these effects are associated with caffeine. The paper is well written and the RCT has significant translational potential, but there are major concerns.

 Major comments:

 Page 2 lines 93-96, page 3 lines 127-35, page 4 lines 173-184: Please move some of the experimental design aspects (lines 92-95) to the appropriate methods section. The authors indicate that blood samples were drawn at ~3 hrs after the 2nd supplement dose, but it is stated that energy expenditure was measured continuously in the chambers. It is not clear what was did to the subjects and when. A figure with a time-line showing the experimental interventions should help better understand the design.

 RESPONSE:  Thank you for taking the time and effort to review our paper.

We added statements to the methods sections to clarify the plasma caffeine sampling time point and sample acquisition for rs762551 genotyping. We added some other statements to clarify the study design. We did not add an additional figure for the research design because the current figures (2 and 4) provide sufficient detail in the legends (in addition to what is explained in the methods). We kept the introduction the same because we feel this explains the hypothesis.

Page 4 lines 185-190, and page 5 lines 231-236: Though there is extensive description of the diet, there is no data on food intake and whether the MFC affected any indices of satiety. It is important to provide caloric intake data after each intervention and determine whether the intake is correlated with energy expenditure and other readouts.

RESPONSE:  The same exact diet was fed to participants (both trials) as carefully explained in the methods section.  Thus in this crossover study (with participants acting as their own controls), diet intake was matched (under eucaloric conditions) to avoid confounding. We did not measure satiety because the participants were in eucaloric conditions, and thus the potential impact on energy expenditure is small to nonexistent.

 Page 6 lines 239-253 and Figure 2: Overall, the study seems to lack mechanistic depth into why and how the MFC increased expenditure. For example, the data indicates that the MFC increased total energy expenditure but does not partition this into other components other than showing a lack of any changes in activity. Though Table 1 shows expenditure in different segments it’s not clear to me whether there is a specific diet effect. As the expenditure was monitored after an overnight fast, can you estimate diet-induced thermogenesis? This could provide some insights into the components of expenditure that are altered.

RESPONSE:  We emphasize in the introduction and discussion that the data are consistent with other flavonoid-caffeine studies that the caffeine portion of the supplement was nearly entirely responsible for the increase in energy expenditure relative to placebo. We also emphasize that the flavonoids may have had a small effect on energy expenditure and fat oxidation (and provide literature support). The two trials were matched in every detail for each participant.  Thus the only difference between trials was the MFC supplement (relative to placebo).  We provided Table 1 to compare metabolic outcomes during specific time segments, but did not specifically measure the thermic effect of the meals (not the focus here).

Page 3 lines 127-35, page 9 lines 308-309, and page 10 lines 340-346: A major limitation of the design is the use of a supplement that contains a complex mixture of flavonoids and caffeine and the inability to parse the individual effects. Though the authors briefly touched on this limitation (lines 340-343), I think this needs to be elaborated more. The authors estimate that the caffeine content of their diet would have increased expenditure by ~46 kcal vs the predicted ~69 kcal. This is a much larger ~50% differential but there is no clear discussion on this other than a vague mention of method differences. Majority of the discussion focusses on caffeine but it’s not clear whether the other components in MFC could potentially decrease expenditure and counteract the effects of caffeine.

RESPONSE:  We feel that the introduction and discussion provide the rationale and literature comparison needed for this study evaluating the influence of a mixed flavonoid-caffeine supplement on energy expenditure.  We also emphasize that part of the difference with other studies is our inclusion of female participants compared to other studies that included only males.

Page 10 lines 355-356: The energy expenditure responses to MFC were only measured to 22 h. There is no data provided to indicate that the MFC would lead to sustained elevation in expenditure. In the absence of such data, the claim that the MFC could offer a long-term strategy for obesity is unfounded.

RESPONSE:  We eliminated this statement.

Round 2

Reviewer 2 Report

The last sentence of the discussion referring to the study by Hill is a bit of a 'stand-alone' remark. Please provide more context or delete the sentence. 

Author Response

Deleted this statement as you recommended (and linked reference).

Reviewer 3 Report

General comments:

The authors revised the manuscript and addressed some of my previous concerns. However, there are some issues that remain to be addressed.

Major comments:

Lines 195-203: A key factor determining energy balance is food intake and diet-induced thermogenesis is a crucial component of energy expenditure. The authors argument that “the potential impact on energy expenditure is small to non existent” doesn’t make any sense to me. Even if satiety is not measured, and eucaloric diets were designed, the total caloric intake at each visit should be provided at a minimum to confirm that an eucaloric state was achieved, and also for future reproducibility.

Lines 297-300, 322-327, 366-367: The inability to tease out the interactive effects of the MFC components on expenditure are still a concern. At several instances in the manuscript, and also in the rebuttal, the authors reiterate that the caffeine component of the MFC is the major driver of the increase in expenditure. I indicted before that there is a ~50% differential between the predicted vs the actual increase in expenditure from potentially the caffeine component, and they also show that the plasma caffeine is unrelated to the trial variance in energy expenditure. Unless, I’m missing something here, if the caffeine alone in this study is not strongly associated with expenditure and the flavonoids are a minor contributor, then I don’t get what’s contributing to the increase in expenditure. The discussion does not adequately address these issues.

Author Response

Lines 195-203: A key factor determining energy balance is food intake and diet-induced thermogenesis is a crucial component of energy expenditure. The authors argument that “the potential impact on energy expenditure is small to non existent” doesn’t make any sense to me. Even if satiety is not measured, and eucaloric diets were designed, the total caloric intake at each visit should be provided at a minimum to confirm that an eucaloric state was achieved, and also for future reproducibility.

RESPONSE:  The total caloric intake MATCHED the energy expenditure. As explained in the methods, "A baseline menu for each subject was prepared based on an estimated energy expenditure (RMR x 1.3), and then modified (snack and dinner) according to measured energy expenditure data at 7 hours." Notice that we emphasize the adjustments were made to "MEASURED" energy expenditure data at 7 hours. Thus diet energy intake is indirectly but accurately assessed by matching to the actual energy expended. This is the same procedure used by other groups using metabolic chambers. This is more accurate than using food volume or weight measures and then using a food composition software program to estimate energy intake.

Lines 297-300, 322-327, 366-367: The inability to tease out the interactive effects of the MFC components on expenditure are still a concern. At several instances in the manuscript, and also in the rebuttal, the authors reiterate that the caffeine component of the MFC is the major driver of the increase in expenditure. I indicted before that there is a ~50% differential between the predicted vs the actual increase in expenditure from potentially the caffeine component, and they also show that the plasma caffeine is unrelated to the trial variance in energy expenditure. Unless, I’m missing something here, if the caffeine alone in this study is not strongly associated with expenditure and the flavonoids are a minor contributor, then I don’t get what’s contributing to the increase in expenditure. The discussion does not adequately address these issues.

RESPONSE: In the discussion (2nd paragraph) we state:

"The trial difference in energy expenditure (3%) given the MFC supplement dosing regimen is similar to what has been reported by other studies (2% to 8%) using metabolic chambers [7-9]. Our MFC supplement provided caffeine and EGCG amounts that were on the lower end of what has been used in previous studies [7-10]. Rudelle et al. [10] proposed that the increase in energy expenditure with caffeine-catechin supplements is related more to caffeine than the flavonoids.  The formula provided by Rudelle et al. [10], projected an increased energy expenditure of 69 kcal/day for the 214 mg caffeine dose provided in this experiment. Our trial difference of 46 kcal/22 h is slightly below this level, and the smaller effect may in part be related to the female gender of the participants and to differences in study methodologies."

Note that we state the difference may be related to female gender and study methods.  We cannot speculate beyond this.  The female gender (thus lower today body weights than male participants) could be the primary factor here.